# IntSR: An Integrated Generative Framework for Search and Recommendation

## Abstract

Generative recommendation has emerged as a promising paradigm, demonstrating remarkable results in both academic benchmarks and industrial applications. However, existing systems predominantly focus on unifying retrieval and ranking while neglecting the integration of search and recommendation (S&R) tasks. What makes search and recommendation different is how queries are formed: search uses explicit user requests, while recommendation relies on implicit user interests. As for retrieval versus ranking, the distinction comes down to whether the queries are the target items themselves. Recognizing the query as central element, we propose IntSR, an integrated generative framework for S&R. IntSR integrates these disparate tasks using distinct query modalities. It also addresses the increased computational complexity associated with integrated S&R behaviors and the erroneous pattern learning introduced by a dynamically changing corpus. IntSR has been successfully deployed across various scenarios on a large internet platform serving hundreds of millions of users, leading to substantial improvements: +9.34% GMV, +2.76% CTR, and +7.04% ACC in three distinct scenarios.

## 1 Introduction

Search and recommendation (S&R) services are now commonly provided by online platforms, such as YouTube and Amazon. These two tasks operate on shared users and items, creating a natural foundation for the joint modeling and application of S&R. A unified S&R model can better capture user preferences and enhance the effectiveness of both tasks, while also reducing engineering overhead (the left side of Fig. 1). Most of the existing studies on unified S&R modeling are based on traditional deep learning frameworks (Yao et al., 2021; Zhao et al., 2022; Xie et al., 2024).

Despite reliance on extensive human-engineered feature sets and training with massive data volumes, the majority of industrial deep learning based frameworks demonstrate poor computational scalability (Zhao et al., 2023; Zhai et al., 2024). Inspired by the development of Large Language Models (LLMs), the generative framework has become an effective method in search or recommendation systems (Zhai et al., 2024; Chen et al., 2025). Integrating S&R into a single generative framework is a promising paradigm, as it resolves scalability challenges, unifies retrieval and ranking, and leverages joint S&R optimization benefits. However, this problem remains underexplored.

Building such a unified framework primarily faces three key challenges. The first involves unifying search, recommendation, retrieval, and ranking processes in one model. The second addresses designing a module to reduce the computational requirements for autoregressive training when all behaviors are aggregated. The third concerns effective negative sampling to prevent temporal misalignment during extended training periods.

To this end, we first unify S&R tasks, along with their retrieval and ranking processes, within a generative autoregressive framework. To address the first two challenges, we observed that the fundamental difference between S&R lies in how user intent is conveyed: explicitly via queries for search, and implicitly through user interactions for recommendation. Motivated by this, we propose IntSR, a unified framework that formulates both tasks and their retrieval and ranking sub-tasks as conditional generation problems. To further reduce training complexity, we designed a query-driven decoder utilizing Key-Value (KV) cache and separate attention calculations for query placeholders.

Figure 1: S&R systems operate with shared users and items, thus user behaviors and model can be unified. Temporal availability of items should be considered.

Regarding the third challenge, we found that it is primarily due to temporal misalignment of vocabularies. Diverse negative sampling strategies have been proposed and examined across diverse domains and tasks. Examples include random negative sampling (RNS), popularity-based negative sampling (PNS, Mikolov et al. 2013), and hard negative sampling (HNS, Zhang et al. 2013, Lai et al. 2024), etc. However, existing approaches typically fail to address item lifecycle dynamics (the right side of Fig. 1). To address this problem, we propose applying a temporal alignment strategy to existing negative sampling methods, which yields significant performance gains.

The effectiveness of the proposed model is confirmed across two public S&R datasets. Concurrently, the temporal alignment strategy is validated using a proprietary industrial dataset. IntSR has been deployed into the production system, serving hundreds of millions of daily active users. Several of its core components have been fully operational at scale for over six months.

To summarize, our key contributions are threefold:

- **Unification of S&R.** We propose an integrated generative framework for both S&R, where tasks are conditioned by different modalities of the queries. This allows to serve diverse scenarios and tasks with one model.
- **Time-varying vocabulary alignment.** We formally define and address the problem of temporal vocabulary misalignment in autoregression models. Our approach offers considerable performance augmentation to all three existing mainstream sampling methods.
- **Offline demonstrations and online deployment.** We conducted extensive experiments on both widely-used public datasets and industrial service datasets to demonstrate the effectiveness of IntSR. IntSR has been successfully deployed across multiple S&R scenarios.

## 2 PRELIMINARIES

Assume we have a set of users and items represented by $\mathcal{U}$ and $\mathcal{I}$, respectively, the interactions between users and items are denoted by $\mathcal{A}$ (see Appendix A for full notations). User behavioral patterns are highly dependent on their temporal and spatial contexts. $\mathcal{S}$ denote the set of discrete spatiotemporal tokens. $\mathcal{F}$ is the set of user feedback types. For each user $u \in \mathcal{U}$, $\mathcal{A}_u = [(s_v, i_v, a_v)|s_v \in \mathcal{S}, i_v \in \mathcal{I}, a_v \in \mathcal{F}, v \in \{1, 2, ..., n\}]$ denotes the interaction sequence in chronological order. $n$ is the number of interacted items. We show that both recommendation and search along with their underlying retrieval and ranking sub-tasks can be modeled as a conditional generation problem. The objective of the sequential model is to predict the conditional probability distribution with different conditions expressed by queries:

$$P_{retr}^{rec} = P(i_{n+1}|\mathcal{A}_u, s_{n+1}) \tag{1}$$
$$P_{rank}^{rec} = P(a_{n+1}|\mathcal{A}_u, s_{n+1}, i_{n+1}) \tag{2}$$
$$P_{retr}^{src} = P(i_{n+1}|\mathcal{A}_u, s_{n+1}, q_{n+1}) \tag{3}$$
$$P_{rank}^{src} = P(a_{n+1}|\mathcal{A}_u, s_{n+1}, i_{n+1}, q_{n+1}) \tag{4}$$

where $P_{retr}^{rec}$, $P_{rank}^{rec}$, $P_{retr}^{src}$, and $P_{rank}^{src}$ denote the conditional probability for retrieval in recommendation, ranking in recommendation, retrieval in search, and ranking in search, respectively. $a_{n+1}$ is the action user may execute on $i_{n+1}$ and $q_{n+1}$ denotes the query expressing user's current interests.

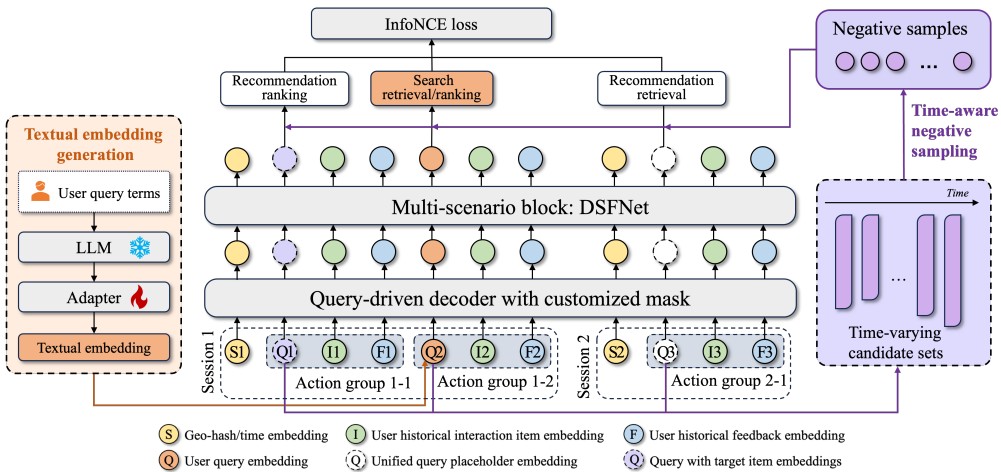

Figure 2: IntSR framework. IntSR unifies different sub-tasks by query types: ranking with candidates which contains multiple items (Q1), and search with natural language queries (Q2). Item online/offline status is incorporated into negative sampling to avoid comparing positive samples with non-existent negatives.

## 3 METHODOLOGY

The overall framework of IntSR is illustrated in Fig. 2. We first present the details of input sequence in Section 3.1. Section 3.2 details how search and recommendation, along with their retrieval and ranking sub-tasks are integrated by query placeholder. When all S&R behaviors are aggregated, Query-Driven Block (QDB) with customized mask is the core module to model user preference and reduce computational complexity (see Section 3.3). DSFNet is used as the multi-scenario block and is detailed in Appendix B. To prevent temporal misalignment during extended training periods, the temporal candidate alignment method is formulated in Section 3.4.

### 3.1 MODELING OF SEQUENCE

The input sequence derived by $\mathcal{A}_u$ comprises four distinct element types, denoted as S, Q, I, and F, respectively. Each element plays a specific role in encoding behavior patterns:

- **S (Scenario tokens).** These represent contextual metadata such as geohash-encoded location tokens or discretized temporal tokens, allowing the model to capture latent user interests associated with specific geographic regions and temporal intervals.

- **Q (Query placeholders).** Functioning as positional markers, Q elements designate locations requiring predictive modeling. Notably, Q should be added only with items that are either involved in the loss computation (e.g., during a specific time step in streaming training) or explicitly searched by the user.

- **I (Item tokens).** Representing items with which users have interacted, positive or negative, these tokens form the core interaction history. In IntSR, item embedding are dense integration of multi-modal information.

- **F (Feedback tokens).** Encoding interaction types such as purchases and clicks, these tokens provide user's feedback to items that informs the model's understanding of user intent and interaction intensity.

### 3.2 UNIFYING SEARCH AND RECOMMENDATION TASKS

In IntSR, the unification of query-free recommendation tasks and query-equipped search tasks is achieved by a general query placeholder Q. As illustrated in Fig. 3, in search tasks, the system is supposed to generate items in response to natural language queries from users, while the information

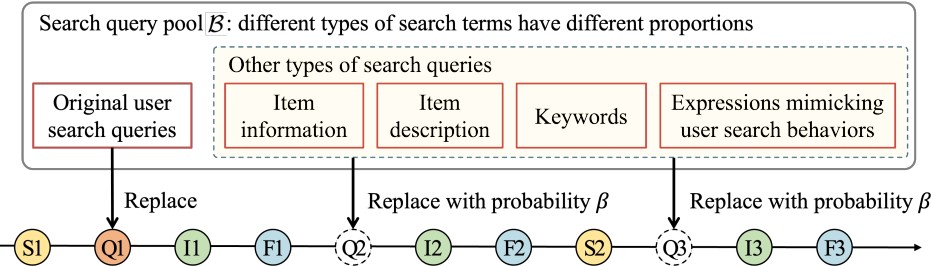

Figure 3: Differences of tasks can be captured by queries. Search task queries contain user-input terms, while ranking task queries include target item information. For recommendation recall, a common query token is used.

of target items should be incorporated in ranking problems. If neither user's explicit query nor item information is integrated, query is replaced by a shared universal token across different users. To convert natural language user search queries into embeddings, we employ a frozen LLM, Qwen3-0.6B (Team, 2025), to generate semantic representations. In search ranking task, this representation is added directly to the embedding of target item or shared query token.

Two strategies are designed to improve generalization of IntSR with respect to natural language queries. The first strategy is for the construction of the query candidate pool. Beyond the original user queries, we also leverage variations generated based on item descriptions and the queries themselves. Specifically, the query pool contains the following types: (1) original user search queries; (2) item information including names, categories, and IP (if applicable); (3) item description and the paraphrased versions of the original description; (4) keywords extracted from (2) and (3); and (5) expressions generated from keywords mimicking user search behaviors (an example in Appendix C).

As illustrated in Fig. 4, the second strategy addresses how the Q positions within the sequence are populated using elements from the aforementioned candidate pool. Let $\mathcal{B}$ denote the query pool constructed above, when a user-item interaction occurs subsequent to a search action, the corresponding Q is populated with actual user queries. For interactions not triggered by a search action, we randomly sample an element from $\mathcal{B}$ and, with a certain probability $\beta$, use it to populate the Q position associated with that interaction.

Figure 4: Integrating search queries to the input sequence. I1: interaction occurs subsequent to a search action. I2 & I3: interactions not triggered by a search action.

### 3.3 QUERY-DRIVEN DECODER WITH CUSTOMIZED MASK

#### 3.3.1 QUERY-DRIVEN BLOCK

We developed QDB based on HSTU (Zhai et al., 2024) for efficient encoding of user histories. QDB separate attention calculations for query placeholders, as expressed by Eqs. (5)-(9), where $X_1$, $X_2$ represent the original sequence and the query placeholder sequence, respectively. The split function partitions the resulting tensor into four components: gating weights $W$, queries $Q$, keys $K$, and values $V$. $Y_1$ and $Y_2$ are the outputs with respect to original sequence $X_1$ and query sequence $X_2$. $A_1$, $M_1$ denotes the attention score and the mask matrix from the original input sequence. $A_{2,k}$, $M_{2,k}$ denotes the attention score and the mask matrix calculated between the query sequence and

the sequence indicated by index $k$. The mask matrix $M$ is derived by three matrices: causal mask, session-wise mask, and invalid Q mask. Positional (Raffel et al., 2020) and ALiBi (Press et al., 2021) temporal relative bias, $\text{rab}_{pos}$ and $\text{rab}_{time}$, are incorporated to refine the initial similarity scores. SiLU (Elfwing et al., 2018) is used as the activation function. $\odot$ denotes Hadamard product.

$$(W_k, Q_k, K_k, V_k) = \text{Split}(\text{SiLU}(\text{MLP}_1(X_k))), k \in \{1, 2\} \tag{5}$$

$$A_1 = M_1 \odot \text{SiLU}(Q_1 K_1^T + \text{rab}_{pos} + \text{rab}_{time}) \tag{6}$$

$$A_{2,k} = M_{2,k} \odot \text{SiLU}(Q_2 K_k^T + \text{rab}_{pos} + \text{rab}_{time}), k \in \{1, 2\} \tag{7}$$

$$Y_1 = \text{MLP}_2(\text{Norm}\,(A_1 V_1) \odot W_1) \tag{8}$$

$$Y_2 = \text{MLP}_2(\text{Norm}\,(A_{2,1} V_1 + A_{2,2} V_2) \odot W_2) \tag{9}$$

Considering a ranking task, this optimization reduces HSTU's computational complexity from $\mathcal{O}(c'N^2)$ to $\mathcal{O}(c'J(N+1))$. $c'$ is candidates per query, $J$ is query placeholder count, and $N$ is the original input sequence length. $J$ primarily accounts for behaviors needing learning in Q within the streaming training time slice, making $J \ll N$, attributable to the superior efficiency of QDB compared to HSTU. Furthermore, similar acceleration gains are achievable if HSTU is replaced by transformer architectures. More implementation details and efficiency experiments are provided in Appendix D.

### 3.3.2 SESSION-WISE MASK AND INVALID Q MASK

To maintain consistency between offline training and online deployment, we propose a session-wise masking mechanism that imposes additional temporal constraints into the encoding of user interaction sequences. As illustrated in Fig. 5, a typical user shopping journey follows the sequence: "browse → click → purchase". Merely applying causal masking makes that the purchase action would inappropriately observe preceding interactions with the same item (see top-left of Fig. 5). To resolve this discrepancy, IntSR introduces the session-wise masking to avoid items within the same session to interact with each other (see Appendix E for an example).

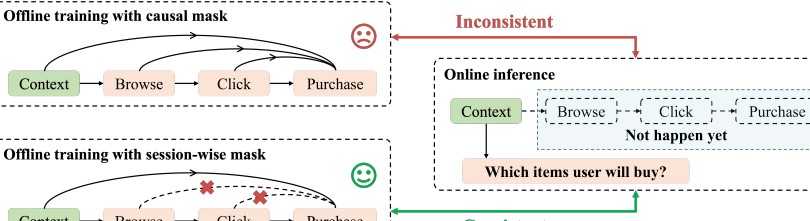

Figure 5: Session-wise masking ensures online-offline consistency. This allows the S&R system to predict item purchases upon page access, even without explicit browsing or clicking.

As previously outlined, Q placeholders accommodate various query types: user search requests, positive/negative target item sets, and a shared universal token. Since Q is part of the input sequence, its representation can influence all tokens. However, Q tokens can only serve as keys and values when encoded as user queries. Invalid Q tokens are explicitly excluded from the attention computation to ensure reasonable final representations (see Appendix E for an example).

### 3.4 SOLVING TIME-VARYING VOCABULARY MISALIGNMENT

As demonstrated in prior discussions, comparison should be grounded in the co-existence of positive and negative samples. This can be achieved by using a loss function with temporal candidate alignment. For IntSR, we use the InfoNCE loss to update model parameters, as expressed by Eq. (10). For each user-item interaction $a \in \mathcal{A}_u$, $i_+$ denotes the ground truth item, and $\mathcal{I}_{t_a} \subseteq \mathcal{I}$ represents the available candidate set at timestamp $t_a$ when interaction $a$ occurs. Let $o_{u,a}$ denotes the output of DSFNet encapsulating the input sequence, $z_{u,a,i} = \text{sim}(o_{u,a}, \text{emb}_i)$ is the score of item $i$.

$\delta_{u,a} \in \{0, 1\}$ is a binary constant that indicates whether the corresponding interaction should be learned by the model.

$$L = -\frac{1}{|\mathcal{A}|} \sum_{u \in \mathcal{U}} \sum_{a \in \mathcal{A}_u} \delta_{u,a} \log \frac{\exp(z_{u,a,i+})}{\sum_{i \in \mathcal{I}_{t_a}} \exp(z_{u,a,i})} \tag{10}$$

Note that calculating Eq. (10) may be computational-expensive under large size of the whole candidate set $\mathcal{I}_{t_a}$. Thus, negative sampling is necessary to improve training efficiency, which should be constrained by the temporal alignment, i.e., only instances that exactly exist when user-item interaction occurs can be treated as negative samples. This can be expressed by Eq. (11), where $\text{prob}_i$ represents the probability of item $i$ being sampled as a negative instance and can be defined according to specific negative sampling strategy. $\mathcal{I}_t$ represents the set of all available candidates at timestamp $t$. The final probability, $\text{prob}_{i,t}$, is determined by both $\text{prob}_i$ and $\mathcal{I}_t$.

$$\text{prob}_{i,t} = \begin{cases} \text{prob}_i, & if \quad i \in \mathcal{I}_t, \\ 0, & otherwise. \end{cases} \tag{11}$$

## 4 EXPERIMENTS

A series of experiments are conducted and reported to answer the following Research Questions:

- **RQ1:** How does proposed IntSR perform on S&R tasks compared with other baselines?
- **RQ2:** To what extent does candidate misalignment impact generative model performance?
- **RQ3:** How does each module in IntSR contribute to its final performance?
- **RQ4:** What is the impact of model width and depth on scaling?

### 4.1 EXPERIMENT SETTINGS

#### 4.1.1 DATASETS AND BASELINES

To evaluate our proposed model, we conduct experiments on a combination of public benchmarks and industrial datasets. Specifically, to answer RQ1 and RQ3, the overall effectiveness of IntSR is assessed on two widely used public datasets that contains both S&R behaviors: KuaiSAR[1] (Sun et al., 2023) and Amazon[2]. We evaluate the effectiveness of candidate alignment on one industrial dataset (RQ2). Its explicit information of item lifecycle allow temporal-aligned sampling and whole-candidate-set evaluation for more convincing performance comparisons. We investigated the impact of model width and depth on scaling (RQ4) using this same industrial dataset. Details of three datasets are provided in Appendix F.

A series of state-of-the-art methods of recommendation, search, and joint models are used as baselines. The recommendation baselines without leveraging search data include the following: (1) DIN (Zhou et al., 2018) captures user interest from historical behaviors using an attention mechanism. (2) SASRec (Kang & McAuley, 2018) is a classic transformer-based sequential recommendation model. (3) BERT4Rec (Sun et al., 2019) is a sequential recommendation model applying a bidirectional transformer. (4) FMLP (Zhou et al., 2022) is an all-MLP sequential recommendation model with feature filtering in frequency domain. (5) HSTU (Zhai et al., 2024) is a autoregressive architecture designed to model user preference.

The baselines for search tasks without using recommendation data include the following: (1) HEM (Ai et al., 2017) learns semantic representations of users, queries and items using a hierarchical embedding model. (2) ZAM (Ai et al., 2019) applies an attention mechanism for history aggregation and controls the personalization degree by a zero attention strategy. (3) TEM (Bi et al., 2020) is a transformer-based embedding model for personalized product search. (4) CoPPS (Dai et al., 2023) applies contrastive learning to learn user representations.

Joint S&R baselines include the following: (1) JSR (Zamani & Croft, 2018) models S&R tasks with a joint loss. (2) USER (Yao et al., 2021) models S&R tasks on an integrated sequence of user

---

[1]https://kuaisar.github.io/

[2]http://jmcauley.ucsd.edu/data/amazon/

behaviors from both domains. (3) UnifiedSSR (Xie et al., 2024) models S&R tasks using a dual-branch architecture with shared parameters and separated behavior sequences. (4) UniSAR (Shi et al., 2024) models the transition behaviors between S&R.

### 4.1.2 IMPLEMENTATION DETAILS

Widely used metrics in S&R systems, top-$k$ Hit Rate (HR@$k$) and Normalized Discounted Cumulative Gain (NDCG@$k$), are employed to evaluate model performance, with $k \in \{1, 5, 10\}$.

Settings of experiments on public datasets are kept as consistent as possible with the open-source code repository released by Shi et al. (2024). When training IntSR, we use 3 QDBs and set embedding size $d$ to 32. The number of historical recommendation and search behaviors visible for each action was fixed at 30 during both training and inference. The learning rate is set to $1 \times 10^{-3}$ and batch size is set to 32. Following previous works, the model performances on public datasets are evaluated on 99 randomly sampled negative instances that user has not interacted with. For KuaiSAR, due to sparse search behaviors after 5-core filtering, we train IntSR with recommendation loss first then fine tune the model with search loss. Since the search behaviors of Amazon (Kindle Store) are repetition of recommendation behaviors, we apply a mask mechanism to avoid label leakage during model training and inference. Implementation details of IntSR on the industrial dataset are provided in Appendix G.

Table 1: Overall performance of IntSR and baselines on search task. * indicates a statistically significant improvement of IntSR over the strongest baseline ($t$-test, $p$-value $< 0.01$).

| Dataset | Model | HR@1 | HR@5 | HR@10 | N@5 | N@10 |
|---|---|---|---|---|---|---|
| Amazon | HEM$^\dagger$ | 0.2497 | 0.6778 | 0.8267 | 0.4736 | 0.5221 |
| | ZAM$^\dagger$ | 0.2954 | 0.7109 | 0.8468 | 0.5147 | 0.5590 |
| | TEM$^\dagger$ | 0.4090 | 0.8185 | **0.9051** | 0.6303 | 0.6587 |
| | CoPPS$^\dagger$ | 0.4052 | 0.8169 | **0.9051** | 0.6281 | 0.6570 |
| | JSR$^\dagger$ | 0.3176 | 0.7038 | 0.8225 | 0.5173 | 0.5563 |
| | USER$^\dagger$ | 0.4123 | 0.7631 | 0.8697 | 0.6000 | 0.6348 |
| | UnifiedSSR$^\dagger$ | 0.3663 | 0.7744 | 0.8812 | 0.5847 | 0.6196 |
| | UniSAR | 0.5343 | 0.8190 | 0.8977 | 0.6875 | 0.7132 |
| | IntSR | **0.5678**$^*$ | **0.8266**$^*$ | 0.8920 | **0.7091**$^*$ | **0.7305**$^*$ |
| KuaiSAR | HEM$^\dagger$ | 0.3337 | 0.6505 | 0.7653 | 0.5029 | 0.5400 |
| | ZAM$^\dagger$ | 0.2815 | 0.6117 | 0.7344 | 0.4560 | 0.4959 |
| | TEM$^\dagger$ | 0.3045 | 0.6502 | 0.7632 | 0.4887 | 0.5254 |
| | CoPPS$^\dagger$ | 0.3117 | 0.6616 | 0.7707 | 0.4977 | 0.5331 |
| | JSR$^\dagger$ | 0.4543 | 0.7162 | 0.7961 | 0.5962 | 0.6221 |
| | USER$^\dagger$ | 0.4628 | 0.7304 | 0.8149 | 0.6069 | 0.6342 |
| | UnifiedSSR$^\dagger$ | 0.4389 | 0.7377 | 0.8320 | 0.5991 | 0.6297 |
| | UniSAR | 0.5282 | 0.7476 | 0.8369 | 0.6417 | 0.6708 |
| | IntSR | **0.5685**$^*$ | **0.7950**$^*$ | **0.8516** | **0.6945**$^*$ | **0.7128**$^*$ |

### 4.2 EFFECTIVENESS OF INTSR IN S&R TASKS (RQ1)

Table 1 and Table 2 provide the results of S&R tasks on two public datasets. We abbreviate NDCG as "N". The best results are in boldface and the second best are underlined, and this convention holds for all other tables. Baselines marked with † mean that the related results are directly reported from their respective papers (Shi et al., 2024). Other values are obtained from our reproduced experiments or our proposed model. IntSR consistently achieves state-of-the-art performance across most evaluation metrics (e.g., HR@1, NDCG@5, NDCG@10) on both the Amazon and KuaiSAR datasets. The model excels in HR@1 and NDCG@5, confirming its enhanced capability to give a high score to the most relevant results. This highlights IntSR's effectiveness and efficiency in search tasks. For recommendation tasks, according to Table 2, IntSR consistently demonstrates superior

Table 2: Overall performance of IntSR and baselines on recommendation task. * indicates a statistically significant improvement of IntSR over the strongest baseline ($t$-test, $p$-value $< 0.01$).

| Dataset | Model | HR@1 | HR@5 | HR@10 | N@5 | N@10 |
|---------|-------|------|------|-------|-----|------|
| Amazon | DIN$^{\dagger}$ | 0.2159 | 0.5170 | 0.6525 | 0.3726 | 0.4165 |
| | SASRec$^{\dagger}$ | 0.2059 | 0.5295 | 0.6772 | 0.3747 | 0.4225 |
| | BERT4Rec$^{\dagger}$ | 0.2481 | 0.5311 | 0.6658 | 0.3954 | 0.4390 |
| | FMLP$^{\dagger}$ | 0.1991 | 0.5356 | 0.6879 | 0.3739 | 0.4232 |
| | HSTU | 0.3446 | 0.6205 | 0.7278 | 0.4908 | 0.5256 |
| | JSR$^{\dagger}$ | 0.2346 | 0.5467 | 0.6779 | 0.3970 | 0.4396 |
| | USER$^{\dagger}$ | 0.2361 | 0.5441 | 0.6854 | 0.3964 | 0.4422 |
| | UnifiedSSR$^{\dagger}$ | 0.2013 | 0.5196 | 0.6707 | 0.3662 | 0.4151 |
| | UniSAR | 0.3010 | 0.5874 | 0.7020 | 0.4513 | 0.4885 |
| | IntSR | **0.3740**$^*$ | **0.6561**$^*$ | **0.7574**$^*$ | **0.5242**$^*$ | **0.5570**$^*$ |
| KuaiSAR | DIN$^{\dagger}$ | 0.1629 | 0.4509 | 0.6179 | 0.3104 | 0.3643 |
| | SASRec$^{\dagger}$ | 0.1249 | 0.4065 | 0.6007 | 0.2671 | 0.3298 |
| | BERT4Rec$^{\dagger}$ | 0.1061 | 0.3699 | 0.5885 | 0.2381 | 0.3083 |
| | FMLP$^{\dagger}$ | 0.1370 | 0.4292 | 0.6159 | 0.2851 | 0.3453 |
| | HSTU | 0.1881 | 0.4920 | 0.6757 | 0.3444 | 0.4037 |
| | JSR$^{\dagger}$ | 0.1754 | 0.4791 | 0.6453 | 0.3315 | 0.3853 |
| | USER$^{\dagger}$ | 0.1489 | 0.4086 | 0.5627 | 0.2820 | 0.3318 |
| | UnifiedSSR$^{\dagger}$ | 0.1225 | 0.3981 | 0.5939 | 0.2617 | 0.3249 |
| | UniSAR | 0.1990 | 0.5169 | 0.6792 | 0.3632 | 0.4158 |
| | IntSR | **0.2179**$^*$ | **0.5373**$^*$ | **0.7248**$^*$ | **0.3815**$^*$ | **0.4421**$^*$ |

performance. Notably, IntSR's impressive performance in HR@1 underscores its exceptional ability to position the most relevant item at the top, which is crucial for effective recommendation systems.

### 4.3 INFLUENCE OF CANDIDATE SET MISMATCH (RQ2)

Table 3: Performance comparison of different negative sampling strategies on industrial dataset.

| Negative sampling strategy | HR@1 | HR@5 | HR@10 | N@5 | N@10 |
|---------------------------|------|------|-------|-----|------|
| RNS | 0.1426 | 0.3691 | 0.4991 | 0.2592 | 0.3012 |
| RNS (aligned) | **0.1810** | **0.4269** | **0.5573** | **0.3075** | **0.3497** |
| PNS (best) | 0.1430 | 0.3655 | 0.4914 | 0.2576 | 0.2983 |
| PNS (best, aligned) | **0.1760** | **0.3949** | **0.5327** | **0.2817** | **0.3264** |
| HNS | 0.1569 | 0.3880 | 0.5150 | 0.2763 | 0.3173 |
| HNS (aligned) | **0.1842** | **0.4305** | **0.5601** | **0.3112** | **0.3533** |

We validate the effectiveness of temporal candidate alignment on the industrial dataset with several popular negative sampling strategies. Instead of the common practice of evaluating the model against the entire set of items, we evaluate it using only the items that were available at the time each user-item interaction occurred. The number of negative samples are set to 20. For hard sampling strategy, we choose 20 items with the highest prediction scores at each training step as negative samples. Results are presented in Table 3. "aligned" indicates that these strategies are enhanced with candidate alignment. For PNS which uses a power coefficient $\alpha$ to control sampling probability based on frequency, we tune $\alpha$ over a range of values and report the best results.

As shown in the Table 3, incorporating our proposed temporal alignment strategy for candidate sets consistently yields substantial performance improvements, regardless of the negative sampling

method employed. Candidate alignment not only improves hit rate but also significantly enhances the ranking quality (NDCG) by placing correct items at more front positions.

## 4.4 ABLATION STUDY (RQ3)

Table 4: Ablation result. For brevity, "session mask" means "session-wise mask". All modules contribute positively to the model's performance. Removing session-wise mask decreases model performance the most. Besides, search queries plays an important role in performance of both task.

| Task | Model | HR@1 | HR@5 | HR@10 | N@5 | N@10 |
|------|-------|------|------|-------|-----|------|
| Search | w/o S | 0.5516 | 0.8169 | 0.8867 | 0.6962 | 0.7189 |
| | w/o search queries | 0.4023 | 0.6453 | 0.7406 | 0.5315 | 0.5624 |
| | w/o session mask | 0.2024 | 0.3958 | 0.5008 | 0.3030 | 0.3369 |
| | w/o DSFNet | 0.5560 | 0.8157 | 0.8836 | 0.6975 | 0.7196 |
| | w/o relative bias | 0.5050 | 0.7861 | 0.8644 | 0.6568 | 0.6823 |
| | IntSR | **0.5678** | **0.8266** | **0.8920** | **0.7091** | **0.7305** |
| Recommendation | w/o S | 0.3311 | 0.6076 | 0.7162 | 0.4779 | 0.5131 |
| | w/o search queries | 0.3325 | 0.6008 | 0.7090 | 0.4746 | 0.5096 |
| | w/o session mask | 0.2864 | 0.5292 | 0.6355 | 0.4142 | 0.4486 |
| | w/o DSFNet | 0.3584 | 0.6329 | 0.7391 | 0.5041 | 0.5386 |
| | w/o relative bias | 0.3574 | 0.6419 | 0.7464 | 0.5082 | 0.5421 |
| | IntSR | **0.3740** | **0.6561** | **0.7574** | **0.5242** | **0.5570** |

Ablation experiments are performed with five variants of IntSR on Amazon to verify the contribution of each components: (1) w/o S: S tokens carrying the spatiotemporal information (only temporal information in public datasets) is removed in the input sequence; (2) w/o search queries: search queries are removed; (3) w/o session-wise mask: only causal mask and invalid Q mask are applied in self-attention calculation of qeury-driven block; (4) w/o DSFNet: DSFNet module is replaced by MLPs; and (5) w/o relative bias: both relative positional and temporal bias in QDB are removed.

Table 4 shows the results on both tasks. The experimental results demonstrate a positive contribution from every module to the model's performance. As mentioned above, search behaviors in Amazon dataset is the duplication of recommendation behaviors, therefore, we can define sessions according to each pair of duplicated behaviors and employ session-wise mask. It is indicated that session-wise mask improves model performance the most, since it prohibits the model focus on user interests rather than the immediate preceding interactions. The results of w/o search queries highlight the advantage of jointly modeling search and recommendation tasks: utilizing search queries improves the recommendation performance.

## 4.5 SCALING-LAW VALIDATION (RQ4)

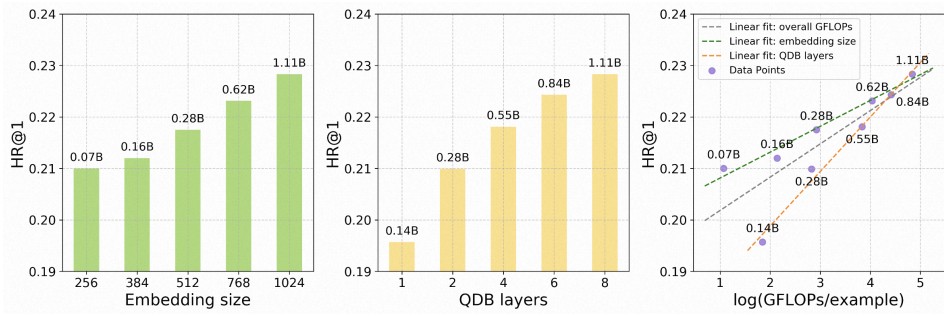

Figure 6: Scaling results of IntSR. Scaling up the model by adjusting the embedding size and the number of QDB layers leads to continuous performance improvement.

We scaled up IntSR by adjusting the embedding size and the number of QDB layers on one of our industrial dataset. The model's parameter size was varied from 0.07B to 1.11B. Results are

presented in Fig. 6. The left subplot shows the effect of various embedding size on HR@1, with the number of QDB layers fixed at 8. The middle subplot shows the effect of varying the number of QDB layers, with the embedding size fixed at 1024. The right subplot illustrates the relationship between HR@1 and FLOPs. The results demonstrate a consistent improvement in HR@1 as the model size increases, which confirms the strong scalability of IntSR. Specifically, a relative increase of 14.3% is observed when the model parameter size is scaled up from 0.14B to 1.11B.

Additionally, it is observed that the fitted line for the QDB layers has a steeper slope compared to that for the embedding size. This indicates that, under conditions where embedding size and depth are no longer limiting factors, adding layers leads to larger performance gains.

### 4.6 ONLINE A/B TEST (RQ1)

We conduct online A/B experiments in three product scenarios to validate the effectiveness of IntSR. For the control group, we randomly selected 10% of users and routed their requests to the production baseline model. In the first scenario, IntSR has achieved a 9.34% relative increase in the overall Gross Merchandise Volume (GMV). IntSR also achieves a 2.76% relative lift in Click Through Rate (CTR) for the second scenario and improves accuracy (ACC) by 7.04% for the third scenario.

## 5 RELATED WORKS

**Generative Recommendation.** The recent success of LLMs has inspired a growing interest in adopting generative frameworks for recommendation and search tasks (Rajput et al., 2023; Zhai et al., 2024; Chen et al., 2025). These efforts can be categorized into two main paradigms. The first paradigm leverages LLMs as a direct predictors (Wu et al., 2024). Typically, user's historical interaction sequences and profile features are converted into textual inputs (often via task-specific prompts). LLMs are expected to generate recommendation results based on the inputs. The second paradigm focuses on utilizing LLMs as feature extractors and reformulating the recommendation or search task itself into an autoregressive framework, thereby adapting LLM architectures and knowledge to the recommendation or search domain (Zhai et al., 2024; Deng et al., 2025).

**Joint Search and Recommendation.** The integration of S&R has emerged as a significant trend in recent years. One approach focuses on search-enhanced recommendation, where search data is utilized as supplementary input to improve the quality of recommendations (Si et al., 2023a;b). The second category involves unified S&R, which aims for a more holistic joint learning process that simultaneously enhances model performance in both S&R (Zhao et al., 2022; Xie et al., 2024). As generative frameworks are independently used in search and recommendation, the integration of LLMs as direct predictors for joint search and recommendation has commenced (Shi et al., 2025; Zhao et al., 2025). However, these methods present significant implementation challenges in scenarios that require large-scale deployment and low-latency responses. Applying an autoregressive framework within joint search and recommendation is an important task.

**Negative sampling.** Negative sampling refers to the strategy that samples several items from unlabeled data as negative instances. RNS is easy to implement and has been widely employed across diverse recommendation models and tasks (He et al., 2020; Yang et al., 2022). Unlike RNS adopts a uniform sampling probability, PNS selects negative instances according to the popularity (Mikolov et al., 2013; Caselles-Dupré et al., 2018). HNS chooses items that are most likely to be confused with positive samples as negative instances (Huang et al., 2021; Lai et al., 2024).

## 6 CONCLUSION REMARKS

This study presents IntSR, a novel framework that successfully unifies the traditionally separate tasks of recommendation, search, retrieval, and ranking under a single generative paradigm. Our core insight is that these tasks can be elegantly unified by treating the query as the central, distinguishing element. Additionally, the time-varying vocabulary misalignment problem is first identified and formulated. We demonstrated that failing to account for the dynamic nature of candidate sets over time leads to erroneous pattern learning. Negative sampling with a dynamic corpus is proposed to address this critical issue. The successful large-scale online deployment of IntSR, yielding state-of-the-art online metrics including substantial increases in CTR, ACC, and GMV.

## ETHICS STATEMENT

Experiments in this study were conducted on two types of datasets: two publicly available datasets and one proprietary industrial dataset. The public datasets are openly accessible, have undergone anonymization, and are broadly employed in prior research. For the proprietary industrial dataset, the collection of all data was authorized by users through their consent obtained during the utilization of the respective software products. All user-related information was anonymized to protect personal privacy, ensuring that researchers could not identify or locate any user-specific private information from the data. We maintained academic integrity throughout all experiments. We will openly share our industrial dataset (fully or partially) and code with respect to temporal-aligned negative sampling, thereby facilitating ethical review and valuable community dialogue.

## REPRODUCIBILITY STATEMENT

To ensure the full reproducibility of the findings presented in this manuscript, we have made comprehensive efforts to document and share the necessary components. The detailed architecture and implementation of our proposed IntSR are thoroughly described and illustrated in Section 3, Appendix B, D, and E. Our experiments utilize tow public datasets and one proprietary industrial dataset. The acquisition, preprocessing steps, and data splitting strategies of public datasets are described in Appendix F. For the private dataset, we provide the statistic information in Appendix F and are going to make it open-access (fully or partially). Implementation details of all experiments are provided in Section 4.1.2. We will make publicly available the code for our temporal-aligned negative sampling strategy to advance studies concerning dynamic vocabulary alignment.

## USE OF LLMS

In the preparation of this manuscript, Large Language Models (LLMs) were utilized solely as a general-purpose writing assistant. Their application was strictly limited to language refinement, correcting spelling and grammatical errors, and enhancing the overall fluidity and clarity of expression. It is crucial to emphasize that all core scientific content, intellectual contributions, and novel ideas presented in this manuscript were exclusively conceived, developed, and verified by the human authors, independent of any LLM involvement. This includes, but is not limited to, the definition of the research problem, the comprehensive literature review, the conceptualization and implementation of the core methodology of IntSR for problem-solving, all content within figures and tables, the collection of all data, the design and execution of experiments, and the subsequent analysis of results. The LLM served purely as a linguistic aid and was not involved in any conceptual or analytical aspect of this research. The authors take full responsibility for the content presented herein.

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

# A NOTATIONS

This appendix provides the meanings of notations used in this study, see Table 5.

Table 5: Notations.

| Symbol | Description |
|---|---|
| $\mathcal{U}$ | Set of all users; the elements in the set are denoted by $u$ |
| $\mathcal{I}$ | Set of all items; the elements in the set are denoted by $i$ |
| $\mathcal{I}_t$ | Set of available items at timestamp $t$; $\mathcal{I}_t \subseteq \mathcal{I}$ |
| $\mathcal{A}$ | Set of all user-item interactions |
| $\mathcal{A}_u$ | Interaction sequence of user $u \in \mathcal{U}$ |
| $\mathcal{S}$ | Set of geo-hash and temporal tokens; the elements in the set are denoted by $s$ |
| $\mathcal{B}$ | Query set containing both original and LLM-generated queries |
| $\mathcal{C}_j$ | Candidate set with respect to the $j_{th}$ query token of input sequence |
| $q_{n+1}$ | Query of user $u$ expressing user's interests of the $(n+1)_{th}$ interaction |
| $p_{n+1}$ | Page context features of the $(n+1)_{th}$ interaction |
| $b_{n+1}$ | Task tag of the $(n+1)_{th}$ interaction to indicate search or recommendation |
| $L$ | Loss function |
| $\delta_{u,a}$ | Binary constant; $\delta_{u,a} = 1$ indicates the corresponding interaction should be learned by the model (prediction loss is contained); otherwise, $\delta_{u,a} = 0$ |
| $o_{u,a}$ | Output of DSFNet related to user $u$ and interaction $a$ |
| $\text{emb}_i$ | Embedding vector of item $i$ |
| $z_{u,a,i}$ | Similarity of $\text{emb}_i$ and $o_{u,a}$; $z_{u,a,i} = \text{sim}(o_{u,a}, \text{emb}_i)$ |
| $\text{prob}_i$ | Probability of sampling item $i$ as negative (without candidate alignment) |
| $\text{prob}_{i,t}$ | Probability of sampling item $i$ as negative at $t$ (with candidate alignment) |
| $X$ | Input features of QDB; $X_1$ is the original sequence; $X_2$ is the query sequence |
| $Q, K, V$ | Query, key, and value matrix before self-attention calculation; when a subscript $k$ is used, $k = 1$ refers to the original input sequence, and $k = 2$ refers to the query sequence |
| $M$ | Mask matrix for attention scores in QDB; calculated as the Hadamard product of causal mask $M_c$, session-wise mask $M_s$, and invalid Q mask $M_Q$; $M_{2,k}$ denotes the mask between the query sequence and the sequence indicated by index $k$ |
| $A$ | Attention scores in QDB; $A_{2,k}$ denotes the attention score calculated between the query sequence and the sequence indicated by index $k$ |
| $\text{rab}_{pos}$ | Relative positional bias |
| $\text{rab}_{time}$ | Relative temporal bias |
| $W$ | Attention output gating weights |
| $Y$ | Output of QDB |
| $f$ | User profile features |
| $R$ | Scenario features; is the combination of $f$, $p^u_{v+1}$, and request features |
| $h$ | Number of query-driven block layers |
| $N$ | Length of input sequence consiting of S, Q, I, F tokens |
| $N_g$ | The number of scenarios defined in DSFNet |
| $d$ | Dimension of embedding space |
| $X_{DSF}, \tilde{X}_{DSF}$ | Input features of DSFNet before and after scenario-aware feature filtering |
| $\gamma_{g,l}$ | Multi-scenario weights in $l_{th}$ DSFNet layer for scenario $g \in \{1, 2, ..., N_g\}$ |
| $\tilde{W}_{g,l}, \tilde{b}_{g,l}$ | Learnable parameters of scenario $g$ and $l_{th}$ DSFNet layer |
| $W_l, b_l$ | Parameters of $l_{th}$ DSFNet layer; equals the weighted sum of $\tilde{W}_{g,l}, \tilde{b}_{g,l}$ |
| $c, c'$ | The number of negative samples and candidates per query, respectively |
| $\beta$ | Replacement probability of non-search Q tokens |

# B DSFNET FOR MULTI-SCENARIO MODELING

Users' behaviors are highly correlated with spatiotemporal context: they exhibit different preferences across various scenarios. These scenarios are formed by combining spatiotemporal features, user's current page context, search or recommendation tag, and personalized user profiles. To address this multi-scenario problem, we employ DSFNet (Yu et al., 2025) after QDB. $N_g$ is a hyperparameter representing the number of scenarios. For each scenario $g \in \{1, 2, ..., N_g\}$, the multi-scenario weights in $l_{th}$ layer, $\gamma_{g,l}$, are derived from the spatiotemporal information $s_{n+1}$, page context $p_{n+1}$, task tag $b_{n+1}$, and user profiles $f$:

$$R = \text{concat}(s_{n+1}, p_{n+1}, b_{n+1}, f) \tag{12}$$

$$\gamma_{g,l} = 2 * \sigma \left( \text{MLP}_{g,l}(R) \right) \tag{13}$$

where $\sigma(\cdot)$ is sigmoid activation function. The factor of 2 allows the weights to exceed 1, enabling feature amplification. The dynamic parameters of $l_{th}$ layer, $W_l$ and $b_l$, are calculated as the weighted sum of all scenarios, as expressed by Eq. (14). $\tilde{W}_{g,l}$ and $\tilde{b}_{g,l}$ are learnable parameters of scenario $g$ and $l_{th}$ layer. Moreover, the scenario information $R$ is used to perform scenario-aware feature filtering on the input feature $X_{DSF}$ before it is passed to the DSFNet block. This is formulated in Eq. (15), where $\tilde{X}_{DSF}$ is features after filtering.

$$W_l = \sum_{g=1}^{N_g} \gamma_{g,l} \tilde{W}_{g,l}, b_l = \sum_{g=1}^{N_g} \gamma_{g,l} \tilde{b}_{g,l} \tag{14}$$

$$\tilde{X}_{DSF} = X_{DSF} \odot \sigma \left( \text{MLP}_3(\text{concat}(X_{DSF}, R)) \right) \tag{15}$$

# C SEARCH QUERY GENERATION

We give an example of item Hello Kitty:

- **Original user search queries:**
  - Hello Kitty
  - Cartoon
- **Item information:**
  - Name: Hello Kitty
  - IP: Hello Kitty
  - Category: Anime
- **Item description:**
  - An iconic, mouth-less white kitten featuring a signature red bow on her head, round eyes, and a pink nose. Her design is simple and soft.
  - Characterized as innocent, kind-hearted, quiet, and friendly, she embodies pure joy "without negative emotions". Her dialogue style is warm, sweet, and adorable.
- **Keywords:** Hello Kitty, Anime, cartoon, kind, quiet, friendly.
- **Expressions mimicking user search behaviors:**
  - Recommend some Hello Kitty items for me.
  - Any recommendations for Anime?

Fig. 4 depicts how the search queries are integrated into the input sequence. In addition to original user submissions, four other types of queries are incorporated into the input sequence with a predefined probability, a method that significantly improves the model's generalization and robustness.

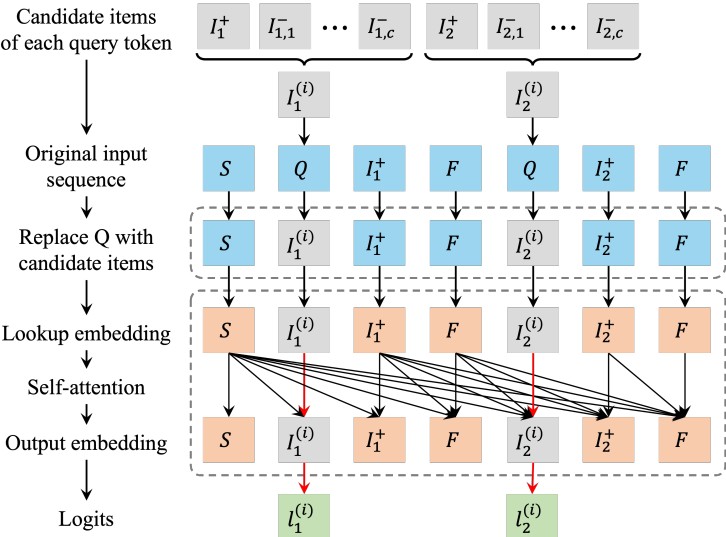

Figure 7: An Ranking Query Example. Each query token is replaced with candidate item tokens for logit prediction. In the attention operation, the candidate tokens can only attend to themselves, as indicated by the red arrows.

# D    IMPLEMENTATION DETAILS OF QUERY-DRIVEN DECODER

## D.1    RANKING QUERY EXAMPLE

A query token can be a user query, a unified token, some item representations, or a mix of these. As illustrated in Fig. 7, using the recommendation ranking task as an example, query-driven decoder aims to predict the probability of query tokens at specific positions marked by query placeholders. These predictions provide ranking scores for each candidate item.

Each group of candidates consists of one positive and multiple negative samples. During training, each query token $q_j$ (a sequence may contain multiple such placeholders) is replaced with its corresponding candidate item token $I_{j,i}$, where $j \in \{1, 2, ..., J\}$ is the $j_{th}$ query token and $J$ represents the total number of query tokens in the input sequence. The modified sequence is input to IntSR and the output is converted to logits of each candidate $z_{a,i}$ by a MLP. At inference time, the query placeholder is appended to the sequence, and the ranking results is determined by the output logits.

## D.2    EFFICIENT CANDIDATE LOGIT COMPUTATION

Direct implementation of HSTU introduces significant computational overhead. Specifically, if we denote the number of negative samples per query as $c$, the computational cost of the ranking model, measured in GFLOPs, becomes $c' = c + 1$ times that of the retrieval model. To mitigate this inefficiency, we adopt a tow-stage computation as shown in Fig. 8.

The first stage processes the original sequence via self-attention and caches the resulting KV-Cache pairs from each layer. In the second stage, candidate embeddings are appended to the original sequence and efficiently processed through the self-attention layers by leveraging the pre-computed KV-Cache. For sequences with multiple query placeholders, the corresponding candidate groups are concatenated sequentially and masked according to Section 3.3.

Let $N$ denote the length of original input sequence, since we transfer repeated computation on the whole sequence into appending candidates to the sequence, the attention mask is thereby enlarged from $N \times N$ to $(N+C) \times (N+C)$ where $C = \sum_{j=1}^{J} |\mathcal{C}_j| = Jc'$, where $\mathcal{C}_j$ is the candidate set with respect to $j_{th}$ query token, including both positive and negative instances. As shown in Fig. 9, the expanded attention matrix is constructed by following steps: (1) the left-up $N \times N$ block is identity to original attention mask; (2) the bottom-right part is an identity matrix of size $Jc' \times Jc'$, as the

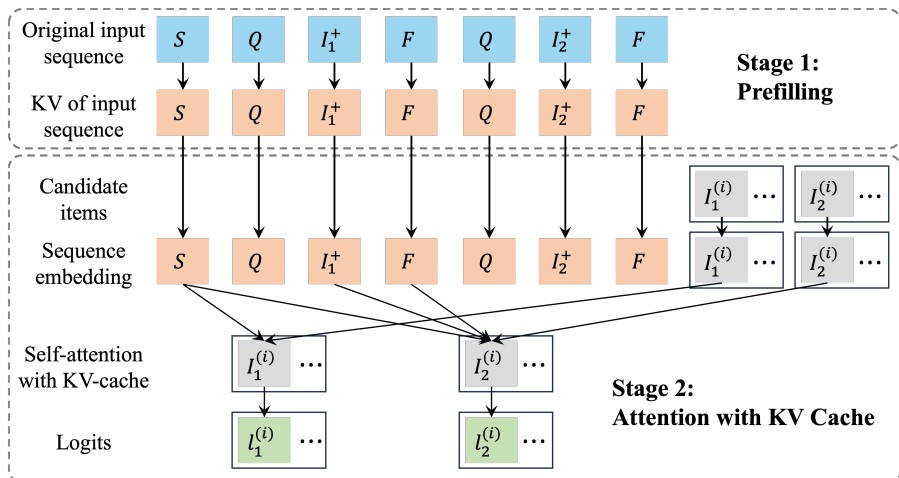

Figure 8: Efficient candidate logit computation with KV-Cache. Initially, the original sequence is encoded by the HSTU (not shown) to compute the keys and values for each token in every HSTU layer. Subsequently, candidate embeddings are computed by applying self-attention using the cached keys and values from the original sequence.

candidate tokens cannot attend to other tokens except for themselves; (3) the top-right part contains $J$ blocks with dimension of $N \times |\mathcal{C}_j|$ and is set to all zeros to prevent candidates from attending to original tokens; and (4) the bottom-left block, comprising $J$ sub-blocks of size $|\mathcal{C}_j| \times N$. To create each sub-block in step (4), we locate the self-attention row corresponding to $j_{th}$ query token within the top-left matrix and replicate it $|\mathcal{C}_j|$ times. As our goal is to compute outputs only for the candidates, the initial $N$ rows of the attention output are omitted. This leaves the last $Jc'$ rows as the final result.

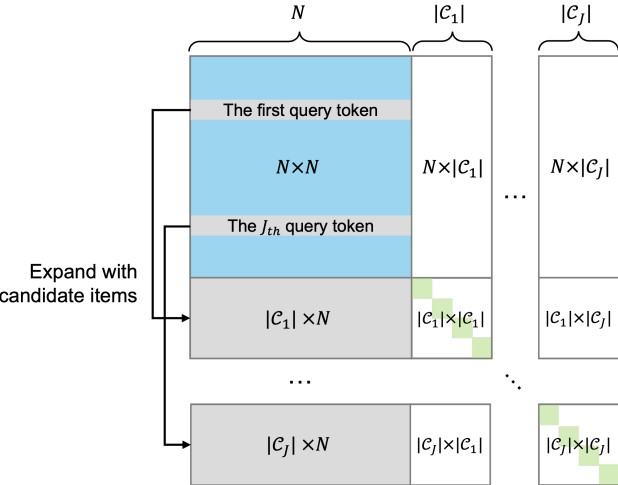

Figure 9: Expanded mask for efficient candidate logit prediction. Matrix dimensions are annotated. White regions indicate zero values. Gray stripes denote single rows, and green squares represent individual elements.

The above optimization reduces the total computational complexity from $\mathcal{O}(c'N^2)$ to $\mathcal{O}(c'J(N + c'J))$. By solving the corresponding quadratic inequality, we find that the overall complexity is reduced when

$$N > \frac{J(1 + \sqrt{1 + 4c'})}{2}.$$

However, when a sequence contains a large number of query tokens (i.e., large $J$), the algorithm becomes less efficient due to the quadratic dependency on $Jc'$. To further improve computational efficiency, we focus on the bottom-right $Jc' \times Jc'$ diagonal block of the attention mask, which governs the interactions among candidate tokens. Most entries in this block are masked out (set to zero), as each candidate token can only attend to itself. An intuitive solution is to decouple the computation of this block from the full attention mechanism, enabling specialized optimization for this structured sparse pattern.

To implement this method, we define (input feature matrix $X$ is omitted here for brevity):

- $Q \in \mathbb{R}^{Jc' \times d}$: query matrix for candidate tokens, where $d$ denotes the dimension of embedding space;

- $[\cdot; \cdot]$: vertical concatenation; $[\cdot, \cdot]$: horizontal concatenation;

- $K = [K_1; K_2], V = [V_1; V_2]$: key and value matrices with divided blocks $K_1, V_1 \in \mathbb{R}^{N \times d}$ and $K_2, V_2 \in \mathbb{R}^{Jc' \times d}$; thus, $K, V \in \mathbb{R}^{(N+Jc') \times d}$;

- $M = [M_1, M_2]$: attention mask with divided blocks $M_1 \in \mathbb{R}^{Jc' \times N}$ and $M_2 \in \mathbb{R}^{Jc' \times Jc'}$; thus, $M \in \mathbb{R}^{Jc' \times (N+Jc')}$.

Under this formulation, the self-attention computation can be equivalently decomposed as Eq (16).

$$
\begin{aligned}
QK^T &= Q[K_1^T, K_2^T] = [QK_1^T, QK_2^T], \\
\text{Attn} &= M \odot (QK^T) = [M_1 \odot QK_1^T, M_2 \odot QK_2^T], \\
\text{Attn}V &= (M_1 \odot QK_1^T)V_1 + (M_2 \odot QK_2^T)V_2
\end{aligned}
\tag{16}
$$

While the term $(M_1 \odot QK_1^T)V_1$ remains challenging to optimize, we observe a key structural property: $M_2 \odot QK_2^T$ is a diagonal matrix (Fig. 9). This implies that only the diagonal entries of $QK_2^T$ need to be computed. Moreover, the result of $(M_2 \odot QK_2^T)V_2$ is equivalent to scaling each row of $V_2$ by the corresponding diagonal element of $M_2 \odot QK_2^T$.

By avoiding the full computation of the $Jc' \times Jc'$ matrix, the complexity of this term is reduced from $\mathcal{O}((Jc')^2)$ to $\mathcal{O}(Jc')$. Because caling the rows of $V_2$ does not change computation complexity, the total computational complexity drops from $\mathcal{O}(Jc'(N + Jc'))$ to $\mathcal{O}(Jc'(N + 1))$. Therefore, the condition for complexity reduction becomes

$$
J < \frac{N^2}{N+1} = N - \frac{N}{N+1} \approx N - 1
$$

This condition, $J < N - 1$, is strictly satisfied, as the input sequence encodes more than just the query tokens.

### D.3 EFFICIENCY EXPERIMENTS

Table 6 provides the comparision experiments on Amazon dataset to measure the efficiency gains of QDB. We compared QDB against its original, un-optimized version within the same IntSR model architecture. Both inference latency (average time to process a single instance) and throughput (instances processed per second) are measured on the same hardware environment (a single NVIDIA H20 GPU with 96 GB memory) with a fixed batch size 64.

Table 6: Performance of QDB in improving latency and throughput

| Model version | Training throughput (instances/sec) ↑ | Inference latency (ms/instance) ↓ | Inference throughput (instances/sec) ↑ |
|---|---|---|---|
| Original module | 21 (1.0×) | 22.5 (1.0×) | 85 (1.0×) |
| QDB | 100 (4.76×) | 18.5 (0.82×) | 160 (1.88×) |

According to Table 6, QDB increases training throughput by a factor of 4.76x, and increases inference throughput by 1.88x while reducing per-instance latency by 18%. The speed-up during training

is more pronounced than during inference. This is because training requires both a forward pass and a computationally expensive backward pass to calculate gradients and update parameters.

As analyzed above, the acceleration effect of QDB is directly correlated with the proportion of query tokens in the input sequence. In our industrial application, we employ a stream-training paradigm where query tokens are assigned only to the newly added interactions in a user's history. This leads to a much lower overall ratio of query tokens. For example, in the industrial dataset we mentioned above, the daily volume of new user interactions accounts for less than 1% of the total dataset size. Therefore, the performance gains from QDB are more significant.

## E    AN EXAMPLE OF CUSTOMIZED MASK

Figure 10: An example of customized masking mechanism. Rows are queries and columns are keys.

Fig. 10 illustrates an example of our customized masking mechanism. Taking the input sequence "S1 → Q1 → I1 → F1 → Q2 → I2 → F2 → S2 → Q3 → I3 → F3" in Figure 2 as an example, Figure 10 illustrates our customized masking mechanism on an $N{\times}N$ mask matrix, where rows are queries and columns are keys, and $N$ signifies the length of the input sequence. Visibility (green with number 1) and invisibility (white) are determined by the following three rules:

- Causal masking: all tokens are masked from attending to subsequent positions in the sequence, resulting in the white upper triangle.
- Invalid Q masking: Q1 and Q3, as invalid instances, are made invisible as a key, preventing it from exposing to other tokens.
- Session-wise masking: tokens within the same session are mutually invisible. For example, action Group 1-1 (Q1, I1, F1) and Action Group 1-2 (Q2, I2, F2) cannot attend to each other. Therefore, Q1's attention is restricted to itself and S1, while Q3 can observe the history of the first session (excluding invalid Q1) as it initiates a new session.

## F    DATASET DETAILS

The overall effectiveness of IntSR is assessed on two widely used public datasets that contains both S&R behaviors: (1) KuaiSAR (Sun et al., 2023) is a dataset of authentic S&R user interactions related to short videos. We adopt the same data preprocessing steps as Shi et al. (2024), and use the last day's data as the test set, the data of second last day as valid set, and the remaining data for training. (2) Amazon is a well-known review dataset in recommendation systems. The search queries and behaviors are created synthetically according to Ai et al. (2017). We choose the subset of "Kindle Store" of the 5-core Amazon dataset. Users and items with less than 5 interactions are removed. Following previous works (Shi et al., 2024), we adopt the leave-one-out strategy to construct train, valid and test dataset. Additionally, one industrial dataset is used to evaluate the effectiveness of temporal alignment sampling. Due to preprocessing and filtering, statistics in Table 7 should not be interpreted as a reflection of the true user population or the entire item corpora.

Table 7: Statistics of the datasets

| Dataset | Users | Items | User-item interactions | |
|---|---|---|---|---|
| | | | Mean | Median |
| Amazon (Kindle Store) | 68223 | 61934 | 28 | 15 |
| KuaiSAR | 22700 | 673415 | 218 | 106 |
| Industrial dataset | 52 M | 819 | 12 | 2 |

## G   IMPLEMENTATION DETAILS ON INDUSTRIAL DATASETS

IntSR model on industrial dataset is trained using Adam optimizer (Kingma & Ba, 2014) with learning rate of $1 \times 10^{-4}$ on 8 NVIDIA H20 GPUs with 96 GB memory. For RQ3, we use 3 QDBs, a sequence length of 500, and an embedding dimension of 128 ($h = 3, N = 500, d = 128$). The batch size is set to 64. Additionally, the number of scenarios for DSFNet is fixed at 2 and 3 layers of DSFNet is used for all experiments. For validation of scaling characteristic (RQ4), we adjusted the number of QDB layers $h$ from 1 to 8, embedding size $d$ from 256 to 1024, and used a fixed sequence length $N = 200$.

