# OpenReview forum: "IntSR: An Integrated Generative Framework for Search and Recommendation"
_ICLR.cc/2026/Conference — Submitted to ICLR 2026_

### Official Review · Reviewer_pYA5 · 2025-11-02

**Soundness:** 3
**Presentation:** 3
**Contribution:** 1
**Rating:** 2
**Confidence:** 4

**Summary:**

This paper proposes IntSR, an integrated generative framework for search and recommendation (S&R) that unifies both tasks and their sub-problems (retrieval and ranking). The model is evaluated on two public datasets (Amazon, KuaiSAR) and an industrial dataset. However this evaluation contains several flaws.

**Strengths:**

* The paper is well written and easy to follow.
* The figures help to understand the methodology described.
* Good ablation studies.

**Weaknesses:**

* The mathematical formulation is unclear because some symbolic notations are not introduced in the main body of the article, but in the appendix. The main body of the article should be self-contained.
* Some related works are missing eg :
   * https://arxiv.org/pdf/2504.06714
   * https://arxiv.org/pdf/2504.05730
* The evaluation section is unclear: the reported results appear very similar to those in Table 3 of this paper. While the authors share their own code, they do not provide access to the code of their competitors, which raises the concern that the results might have been copied from that table, with their own model simply added, rather than rerunning all baselines under the same conditions.
* No statistical tests are present.
* Only two datasets are used.
* The evaluation has been carried out with 99 negatives and one positive, despite the paper “Widespread Flaws in Offline Evaluation of Recommender Systems” shows that it is better to use all the items as negatives.

**Questions:**

1. Could you please add another dataset and the statistical tests?
2. Could you please add the evaluation using all the items as negatives?
3. Could you clarify how the model addresses the cold start problem?
4.To convert natural language user search queries into embedding, why did you choose exactly the Qwen 0.6B model for the without trying any other?

---

> ### Author Response · Authors · 2025-11-21
> **[1/3] Official Comment by Authors**
>
> Thank you for the detailed feedback.
>
> **To W1**: We have conducted a thorough check of the entire paper and found that the explanations of A and M with subscript are missing. These corrections have been incorporated into the revised version we have uploaded. We have rewritten lines 214-216.
>
> **To W2**: For the revised version of our paper, we have rewritten lines 463-480, which comprise the related work.
>
> The modeling paradigm of adopting LLMs to recommendation can be categorized into two types. The first category leverages LLMs as a direct predictors (Wu et al., 2024). Typically, user's historical interaction sequences, profile features are converted into textual inputs (often via task-specific prompts). LLMs are expected to generate recommendation results based on the inputs. The studies mentioned by the reviewer fall into this category. However, this approach presents significant implementation challenges in scenarios that require large-scale deployment and low-latency responses (Jia et al., 2025). In contrast, our proposed IntSR follows the another paradigm. This approach focuses on transferring the knowledge and capabilities of LLMs into a specialized recommendation model by using the LLM as a feature extractor and reformulating the task autoregressively.
>
> In our industrial application, which must handle approximately 10,000 user requests per second (QPS) with response times at the millisecond level, we found the second route to be more feasible.
>
> **To W3**:
>
> **On the similarity of baseline results**: The performance numbers for most baseline models in our paper are identical to those reported in the original UniSAR paper (Shi et al., 2024). This was a transparent choice. UniSAR, being the most recent and state-of-the-art model, already conducted a comprehensive comparison against a series of established baselines. Instead of re-implementing and potentially misconfiguring these baseline models, we adopted the practice of directly citing the results from the primary SOTA paper we are comparing against. This is a common convention in fast-moving fields to ensure that the comparison is made against the strongest possible, author-verified version of the baselines. To ensure full transparency, we explicitly marked these cited results with a specific symbol (e.g., an dagger) in our tables and provided a clear explanation of their origin, as stated in our manuscript.
>
> **On rerunning competitors and fairness**: To ensure fair comparison, we took two critical steps:
> - Fisrt, we reran the main competitor, UniSAR. We used the official, publicly available source code provided by its authors. Our reproduced results for UniSAR were consistent with those reported in their paper, which validates that our experimental environment is correctly configured.
> - In addition, we implemented and evaluated an additional strong baseline (HSTU, Zhai et al., 2024) and our own model (IntSR) under settings that are directly comparable to UniSAR. We aligned our configurations with those specified in the open-source code of UniSAR. Specifically, for implementation of HSTU and IntSR, we adopted hyperparameter settings that were as consistent as possible with UniSAR's, including key parameters such as embedding size, length of the visible sequence, batch size, and learning rate. This ensures that all models we ran ourselves were evaluated under maximally consistent settings. These details are also specified in our paper.

---

> ### Author Response · Authors · 2025-11-21
> **[2/3] Official Comment by Authors**
>
> **To W4 & Q1**: We have included the description of statistical significance, lines 294-317 and lines 349-374.
>
> We conducted multiple runs for both our model, IntSR, and the strongest baseline model, using different random seeds for each run. Based on these results, we performed a t-test to assess whether the improvements of IntSR over the baseline are statistically significant. The results, as shown below, confirm that most of the improvements are significant (p < 0.01).
>
> The detailed results are listed in the table below. In each cell, the two numbers separated by a '/' denote the results for IntSR and the strongest baseline, respectively.
>
> | Dataset and task | Metric | Mean | Standard deviation | p-value |
> | :--- | :--- | :--- | :--- | :--- |
> | Amazon, search | HR@1 | 0.5667 / 0.5207 | 0.0071 / 0.0082 | 2e-8 |
> | | HR@5 | 0.8437 / 0.8137 | 0.0070 / 0.0065 | 3e-8 |
> | | HR@10 | 0.9080 / 0.8918 | 0.0057 / 0.0054 | 1e-6 |
> | | NDCG@5 | 0.7180 / 0.6791 | 0.0067 / 0.0074 | 6e-9 |
> | | NDCG@10 | 0.7391 / 0.7046 | 0.0062 / 0.0070 | 1e-8 |
> | KuaiSAR, search | HR@1 | 0.5677 / 0.4992 | 0.0229 / 0.0392 | 5e-4 |
> | | HR@5 | 0.7951 / 0.7480 | 0.0141 / 0.0058 | 8e-7 |
> | | HR@10 | 0.8405 / 0.8392 | 0.0077 / 0.0056 | 0.3244 |
> | | NDCG@5 | 0.6935 / 0.6307 | 0.0140 / 0.0155 | 5e-6 |
> | | NDCG@10 | 0.7083 / 0.6604 | 0.0124 / 0.0160 | 2e-5 |
> | Amazon, recommendation | HR@1 | 0.3645 / 0.3440 | 0.0062 / 0.0130 | 0.0028 |
> | | HR@5 | 0.6552 / 0.6199 | 0.0093 / 0.0111 | 9e-5 |
> | | HR@10 | 0.7617 / 0.7301 | 0.0090 / 0.0087 | 4e-5 |
> | | NDCG@5 | 0.5191 / 0.4902 | 0.0078 / 0.0118 | 3e-4 |
> | | NDCG@10 | 0.5536 / 0.5260 | 0.0075 / 0.0108 | 2e-4 |
> | KuaiSAR, recommendation | HR@1 | 0.2173 / 0.1886 | 0.0138 / 0.0107 | 5e-4 |
> | | HR@5 | 0.5537 / 0.4994 | 0.0132 / 0.0200 | 8e-5 |
> | | HR@10 | 0.7399 / 0.6614 | 0.0183 / 0.0185 | 2e-6 |
> | | NDCG@5 | 0.3901 / 0.3487 | 0.0100 / 0.0158 | 1e-4 |
> | | NDCG@10 | 0.4504 / 0.4015 | 0.0063 / 0.0152 | 8e-6 |
>
> **To W5 & Q1**: These two datasets are common benchmarks in this research area. They are widely used in related works (e.g., Si et al. (2023) and Shi et al. (2024) used both KuaiSAR and Amazon; Wang et al. (2024), Shen et al. (2024), and Shi et al. (2025) adopted KuaiSAR and another private or open-source dataset), including one of the papers mentioned by the reviewer (Zhao et al., 2025). This adoption as a standard testbed ensures that our results are directly comparable to other state-of-the-art methods.
>
> **To W6 & Q2**: We adopted the sampled-based evaluation (1 positive vs. 99 negatives) to ensure a fair and direct comparison with the baseline models and prior state-of-the-art works we compare against (e.g., Si et al., 2023; Shi et al., 2024; Xie et al., 2024; including two studies you mentioned above). These studies report results using this sampled metric protocol. Adhering to this convention allows for a consistent and transparent assessment of our model's relative improvement over the existing literature.
>
> **To Q3**: Our solution is a combination of system-level strategies and model-level capabilities. In our production environment, we address the cold-start problem using a combined approach of "online warm-up and stream-training". Specifically:
>
> - System-level: New items do not start with zero traffic. They enter a "warm-up" phase, where an external traffic allocation system strategically assigns them a small but growing amount of exposure. This ensures that every new item has an opportunity to gather initial user interactions (clicks, purchases, etc.).
> - Model-level: We employ a stream-training setup to continuously update our model with fresh interaction data. The QDB module is the key enabler. As detailed in our paper, QDB significantly reduces the computational complexity of the self-attention mechanism. This efficiency gain makes it feasible to run frequent, low-latency stream-training jobs.
>
> In summary, our solution to the item cold-start problem is not just a feature within the model, but an integrated system. This also addresses the new user cold-start in a similar manner, where their initial sparse interactions are quickly learned by the model to refine subsequent recommendations.

---

> ### Author Response · Authors · 2025-11-21
> **[3/3] Official Comment by Authors**
>
> **To Q4**: Our decision to select a smaller model was driven by the practical need to strike a balance between performance, cost, and online response time.
>
> On one hand, we operate under strict latency constraints. In our production environment, the process of encoding a user's search query and performing model inference is just one part of the whole pipeline with other upstream and downstream stages. This system must handle approximately 10,000 queries per second (QPS), while the end-to-end response time must be controlled at the millisecond level. These  requirements compel us to carefully optimize the latency of every component. Thus, small-scale LLMs are preferred in these scenarios.
>
> On the other hand, unlike LLMs as direct predictors, which typically exhibit an obvious dependency on LLM for core prediction tasks, IntSR uses the LLM only for encoding search queries. This architectural choice grants us significant flexibility in selecting the LLM for practical deployment. We chose the Qwen model because it had already been successfully deployed at scale in other projects. This prior success guaranteed a more predictable implementation cost and significantly reduced potential risks.
>
> Please let us know if any concerns remain unaddressed; we are happy to discuss them.
>
> References:
> - Si, Z., Sun, Z., Zhang, X., Xu, J., Zang, X., Song, Y., ... & Wen, J. R. (2023, July). When search meets recommendation: Learning disentangled search representation for recommendation. In Proceedings of the 46th international ACM SIGIR conference on research and development in information retrieval (pp. 1313-1323).
> - Shi, T., Si, Z., Xu, J., Zhang, X., Zang, X., Zheng, K., ... & Song, Y. (2024, July). UniSAR: Modeling User Transition Behaviors between Search and Recommendation. In Proceedings of the 47th International ACM SIGIR Conference on Research and Development in Information Retrieval (pp. 1029-1039).
> - Xie, J., Liu, S., Cong, G., & Chen, Z. (2024, May). Unifiedssr: A unified framework of sequential search and recommendation. In Proceedings of the ACM Web Conference 2024 (pp. 3410-3419).
> - Wang, Y., Chen, M., Hu, Y., Guo, W., Zhang, Y., Guo, H., ... & Coates, M. (2024, October). Enhancing Click-through Rate Prediction in Recommendation Domain with Search Query Representation. In Proceedings of the 33rd ACM International Conference on Information and Knowledge Management (pp. 2462-2471).
> - Shen, K., Ding, X., Zheng, Z., Gong, Y., Li, Q., Liu, Z., & Zhang, G. (2024). Seminar: Search enhanced multi-modal interest network and approximate retrieval for lifelong sequential recommendation. arXiv preprint arXiv:2407.10714.
> - Shi, T., Qin, W., Yu, W., Zhang, X., He, M., Fan, J., & Xu, J. (2025). Bridging Search and Recommendation through Latent Cross Reasoning. arXiv preprint arXiv:2508.04152.
> - Zhao, J., Wang, W., Xu, C., Chen, X., Ren, Z., & Verberne, S. (2025). Unifying Search and Recommendation: A Generative Paradigm Inspired by Information Theory. arXiv preprint arXiv:2504.06714.
> - Wu, L., Zheng, Z., Qiu, Z., Wang, H., Gu, H., Shen, T., ... & Chen, E. (2024). A survey on large language models for recommendation. World Wide Web, 27(5), 60.
> - Jia et al., 2025. LEARN: Knowledge Adaptation from Large Language Model to Recommendation for Practical Industrial Application. AAAI 25.

---

> ### Author Response · Authors · 2025-11-24
> **Official Comment by Authors**
>
> Dear Reviewer pYA5,
>
> We are writing to ask whether our rebuttal has addressed your concerns. We are happy to discuss any remaining issues.

---

> > ### Comment · Reviewer_pYA5 · 2025-11-24
> > **Thanks for your rebuttal**
> >
> > Dear authors,
> >
> > You have succesfully addressed all my comments. I believe the paper is much more solid but I am afraid, given the limitations regarding the baselines and the unclear architectural structures you presented the paper is not ready for being published, yet.
> >
> > Thanks

---

> > > ### Author Response · Authors · 2025-11-25
> > > **Official Comment by Authors**
> > >
> > > Dear Reviewer pYA5,
> > >
> > > Thanks for your reply. Could you please clarify which part of the architecture is unclear? We are happy to discuss more.

---

> > > ### Author Response · Authors · 2025-11-28
> > > **Official Comment by Authors**
> > >
> > > Dear Reviewer pYA5,
> > >
> > > As for the architecture, IntSR contains four parts and we'd like to explain them briefly:
> > >
> > > First, "**universal query token**" is our unifying principle for integrated search and recommendation modeling:
> > > - Retrieval in recommendation: Unified query token.
> > > - Ranking in recommendation: Target items to be ranked.
> > > - Retrieval in search: The LLM-generated vector of the user's query words.
> > > - Ranking in search: Target items combined with the user's query words.
> > >
> > > Second, **QDB module makes the autoregression efficiently**. Integrating all tasks into a single model causes large computational costs. QDB module separates the query's attention calculation from the original sequence. The KV of the original sequence is cached, and the query only needs to perform attention calculation once with the cached sequence. This significantly reduces the time complexity, allowing the model to be incrementally trained with hundreds of millions of daily user interactions.
> > >
> > > Third, **multi-scenario block (DSFNet) incorporates user profiles and contextual features**. DSFNet captures scenario-specific user interests effectively, especially in applications highly correlated with spatiotemporal context.
> > >
> > > In addition, **temporal-aligned negative sampling strategy yields significant performance gains** by handling product dynamics during modeling training.
> > >
> > > If you have further questions about the model architecture, please contact us at any time.
> > >
> > > Best regards,
> > >
> > > Authors of Submission 2037

---

### Official Review · Reviewer_spp9 · 2025-11-03

**Soundness:** 3
**Presentation:** 3
**Contribution:** 3
**Rating:** 6
**Confidence:** 4

**Summary:**

his paper proposes IntSR, an integrated generative framework that unifies search and recommendation (S&R) tasks along with their retrieval and ranking sub-tasks. The key innovation is treating queries as the central distinguishing element - using explicit natural language queries for search and implicit behavioral patterns for recommendation. The framework introduces a Query-Driven Block (QDB) with customized masking to reduce computational complexity, and addresses temporal vocabulary misalignment through a candidate alignment strategy. The model has been deployed at scale, serving hundreds of millions of users with reported improvements of +9.34% GMV, +2.76% CTR, and +7.04% ACC across three scenarios.

**Strengths:**

--Practical Impact: The successful deployment at scale with significant business metrics improvement demonstrates real-world value beyond academic contributions.


--Comprehensive Framework: The paper addresses multiple aspects - unification, efficiency, and temporal dynamics - in a coherent framework rather than treating them as isolated problems.


--Temporal Alignment Innovation: The identification and solution of temporal vocabulary misalignment is an important contribution that has been overlooked in prior work. Table 3 convincingly shows consistent improvements across all negative sampling strategies.


--Thorough Experimental Validation: The paper includes experiments on public datasets, proprietary data, ablation studies, and online A/B tests, providing multiple angles of validation.


--Computational Efficiency: The detailed complexity analysis and optimization in Appendix D shows careful engineering consideration for practical deployment.

**Weaknesses:**

--Limited Baseline Comparisons: For the temporal alignment experiments (Table 3), only basic negative sampling strategies are compared. No comparison with recent advanced sampling methods or other temporal-aware approaches.


--Unclear Architectural Choices:


  * Why is DSFNet necessary when QDB already handles multi-scenario aspects through query types?
  * The relationship between the 3-layer QDB and DSFNet modules is not well justified
  * The choice of Qwen3-0.6B for query encoding seems arbitrary

--Dataset Limitations:

* KuaiSAR requires separate training for recommendation and search due to sparse search behaviors, suggesting the unified approach may not work well when data is imbalanced
* Amazon dataset uses synthetic search queries, limiting the validity of search task evaluation
* Industrial dataset details are too vague for reproducibility

**Questions:**

-- Query Generation Strategy: In Section 3.2, you mention using probability β to populate non-search Q positions. How sensitive is the model to this hyperparameter? What values were used in experiments?


-- Temporal Alignment Overhead: Does maintaining temporal availability information (It) for all items introduce significant storage/computation overhead in production? How do you handle items with complex availability patterns?


-- Cold Start Performance: How does IntSR perform for new users or items with limited interaction history? The unified approach might exacerbate cold start issues.


-- Search Query Quality: Table 4 shows a dramatic drop when removing search queries for the search task (HR@1: 0.5678→0.4023). Does this suggest the model is overly dependent on explicit queries rather than learning implicit patterns?

---

> ### Author Response · Authors · 2025-11-21
> **[1/3] Official Comment by Authors**
>
> Thank you for the detailed feedback.
>
> **To W1**:
> We would like to clarify that the proposed temporal-alignment strategy is not intended as a stand-alone negative sampling method. Rather, it is designed as an enhancement module that can be integrated with most existing negative sampling technique.
>
> The purpose of Table 3 is to demonstrate this synergistic effect: regardless of the base negative sampling method employed, incorporating our temporal-alignment strategy consistently improves its performance. To illustrate this point, we selected three commonly used negative sampling methods as representative examples. The results confirm that applying temporal-alignment strategy on top of these methods leads to further performance gains.
>
> **To W2-1**:
> As stated in the paper, the input to DSFNet includes not only the task representation for search or recommendation but also user profile features and contextual page features.
>
> The importance of user profile features is well-established. The page features allow for a finer-grained distinction of scenarios, as accessing different pages within the application often signifies distinct user intents. A key design consideration was how to integrate these features effectively. If these diverse features were all consolidated into the query token, it would obscure the token's crucial role to distinguish between search/recommendation and retrieval/ranking tasks. Moreover, overloading a single token with such varied information could lead to information loss. Therefore, we introduced DSFNet to handle these features.
>
> **To W2-2**:
> Our decision to select a smaller model was driven by the practical need to strike a balance between performance, cost, and online response time.
>
> On one hand, we operate under strict latency constraints. In our production environment, the process of encoding a user's search query and performing model inference is just one part of the whole pipeline with other upstream and downstream stages. This system must handle approximately 10,000 queries per second (QPS), while the end-to-end response time must be controlled at the millisecond level. These  requirements compel us to carefully optimize the latency of every component. Thus, small-scale LLMs are preferred in these scenarios.
>
> On the other hand, unlike LLMs as direct predictors, which typically exhibit an obvious dependency on LLM for core prediction tasks, IntSR uses the LLM for encoding search queries. This architectural choice grants us significant flexibility in selecting the LLM for practical deployment. We chose the Qwen model because it had already been successfully deployed at scale in other projects. This prior success guaranteed a more predictable implementation cost and significantly reduced potential risks.
>
> **To Q1**:
> - Regarding Experiments on Public Datasets: The query generation strategy was not applied, because the proportion of new (unseen) search queries in the evaluation sets of these datasets is relatively small (0.38% in Amazon and 1.15% in KuaiSAR). In this situation, the data augmentation approach less necessary.
> - On the Sensitivity and Choice of beta: We share our experience from the industrial deployment where this strategy is crucial.
>     - Conceptual Role of beta: The beta parameter controls the trade-off between generating synthetic queries (to handle data sparsity) and preserving the original user sequence. A higher beta aggressively creates more query-like tokens, which is beneficial in extreme few-shot or zero-shot scenarios. A lower beta is more conservative.
>     - Empirical Choice: In our production environment, beta is typically set to a relatively small value (e.g., in the range of 0.1-0.3). Our empirical findings suggest that the model is not overly sensitive to the precise value of beta within this reasonable range. The key insight is that even a small amount of synthetic query generation provides a significant benefit in low-data regimes. For the industrial application mentioned, we used a value of beta=0.2.
> - Context of Application: This strategy is specifically designed for our industrial use case where IntSR serves as a "function" for an AI agent, a scenario characterized by data sparsity. For search applications with abundant data, the benefits of this strategy would be limited.

---

> ### Author Response · Authors · 2025-11-21
> **[2/3] Official Comment by Authors**
>
> **To Q2**: Maintaining temporal availability information for all items does not introduce significant storage/computation overhead in production. Here's how we made it efficient:
>
> **Storage**: The temporal availability information for each item is stored as a list of disjoint time intervals (e.g., [start_time_1, end_time_1], [start_time_2, end_time_2], ...). For an item that is always available, we only need to store a single interval. For items with complex availability patterns, this representation is far more efficient than storing full availability status at every timestamp. In our production system, the average number of intervals per item is very low (<2). With this storage approach, every 10 million items require less than 1 GB of storage space.
>
> **Computation**: The computation overhead occurs during the negative sampling of training. We need to ensure that we obtain exactly k valid negative samples for each user interaction. In practice, we adopt an oversampling-and-filtering strategy:
> - Oversample: Instead of drawing just k candidates, we first sample α*k candidate negatives (e.g., via random sampling), where α is a oversampling factor (e.g., 1.2-1.5). The oversampling factor α is a tunable hyperparameter. It is set empirically based on the dataset's average item availability rate to ensure the final pool almost always contains at least k items.
> - Filter: For each of these α*k candidates, we check its availability. With the interval-based storage described above, checking an item's availability at a specific timestamp becomes efficient.
> - Select: select k samples from the valid candidate negative samples rerived from the filtering step.
>
> **To Q3**: IntSR does not include a dedicated mechanism specifically for cold-start items and users. While semantic ID-based methods (e.g., Rajput et al., 2023; Singh et al., 2024) have shown advantages for cold-start issues, our own experiments with the Tiger framework (Rajput et al., 2023) did not yield positive gains in our setting. Therefore, in our practical application, we address the cold-start problem using a combined approach of "online warm-up and stream-training". Specifically:
>
> - System-Level: New items do not start with zero traffic. They enter a "warm-up" phase, where an external traffic allocation system strategically assigns them a small but growing amount of exposure. This ensures that every new item has an opportunity to gather initial user interactions (clicks, purchases, etc.).
>
> - Model-Level: We employ a stream-training setup to continuously update our model with fresh interaction data. The QDB module is the key enabler. As detailed in our paper, QDB significantly reduces the computational complexity of the self-attention mechanism. This efficiency gain makes it feasible to run frequent, low-latency stream-training jobs.
>
> **To Q4**: In a search scenario, the user's query provides an explicit and immediate expression of their current intent. This query plays an important role in the correctness of the immediate results. Consequently, its removal naturally leads to a substantial decline in accuracy, as the model loses the most critical signal. However, even after removing the search queries, the model still retained 71% of its original top-1 accuracy. This suggests that the model is capable of learning the user's implicit interests from their historical interactions.

---

> ### Author Response · Authors · 2025-11-21
> **[3/3] Official Comment by Authors**
>
> **To W3**:
>
> **KuaiSAR**:
>
> In KuaiSAR, the search interactions are extremely sparse compared to recommendation interactions, with a ratio of approximately 1%. In this situation, forcing a single joint-training process could lead to suboptimal performance. The model's learning would be heavily skewed towards the recommendation task.
>
> In contrast, Amazon features a 50% proportion of search data. Three scenarios where IntSR has been successfully deployed exhibit diverse data distributions: Scenario 3 involves no search behavior, while Scenarios 1 and 2 have search data proportions of 13% and 21%, respectively. The effectiveness of IntSR has been validated across these datasets.
>
> Faced with extreme data imbalance, how to enable a model to achieve strong performance through joint training alone, without task-specific fine-tuning, is a profound and valuable research question for the future.
>
> **Amazon**:
>
> The method for generating search data on the Amazon dataset follows a well-established practice in the search domain. The search data is not arbitrarily generated; instead, it is synthetically constructed based on user recommendation behaviors and the products' descriptive metadata.
>
> Following the paradigm from Ai et al. (2017), this construction process involves three main steps: (1) extract the full category hierarchy for each item from its product metadata; (2) concatenate the terms from this hierarchy to create a single descriptive string; (3) clean this string by removing stopwords and duplicate words to form the final query. The query construction method has been adopted by Liu et al. (2022), Si et al. (2023), Shi et al. (2024), Zhao et al. (2025), etc.
>
> Furthermore, this construction method offers a unique advantage for our evaluation on the Amazon dataset. Since the search behaviors are derived from the recommendation data, we are able to accurately construct user sessions that contain both search and recommendation interactions. This provides a perfect testbed for validating the effectiveness of our session mask mechanism.
>
> **Industrial dataset**:
>
> To provide the reviewer and readers with a more intuitive understanding of our industrial data, we provide one example below. Please note that some information within this sample has been processed or anonymized to comply with privacy requirements.
> - user_id: 791273b4c
> - his_seq_rec: 1749304912|1|||26694_9089_15&1749304966|3|692a62a58c|70.0|&...
> - his_seq_src: 1749305060|7|680a62a58c||8&1749305060|7|680a62a58c||9&...
> - profile_1: 48
> - profile_2: 1
>
> In this example, "his_seq_rec" and "his_seq_src" are the sequences of user interactions of recommendation and search. Each sequence is composed of several sub-sequences joined by the "&" character, where each sub-sequence represents a single user interaction. Within each sub-sequence of recommendation, fields are separated by "|" and correspond, in order, to: timestamp, action type, item ID, item price, and location ID. Within each sub-sequence of search, fields are separated by "|" and correspond, in order, to: timestamp, action type, item ID, item price, and query word ID.
>
> To enhance reproducibility upon acceptance, we will:
> - Release a standalone, open-source implementation of our code.
> - Seek internal approval to release a portion of the dataset used.
>
> References:
> - Rajput, S. et al. (2023). Recommender systems with generative retrieval. Advances in Neural Information Processing Systems, 36, 10299-10315.
> - Singh, A. et al. (2024, October). Better generalization with semantic ids: A case study in ranking for recommendations. In Proceedings of the 18th ACM Conference on Recommender Systems (pp. 1039-1044).
> - Ai, Q. et al. (2017, August). Learning a hierarchical embedding model for personalized product search. In Proceedings of the 40th International ACM SIGIR Conference on Research and Development in Information Retrieval (pp. 645-654).
> - Liu, J. et al. (2022, April). A category-aware multi-interest model for personalized product search. In Proceedings of the acm web conference 2022 (pp. 360-368).
> - Si, Z. et al. (2023, July). When search meets recommendation: Learning disentangled search representation for recommendation. In Proceedings of the 46th international ACM SIGIR conference on research and development in information retrieval (pp. 1313-1323).
> - Shi, T. et al. (2024, July). UniSAR: Modeling User Transition Behaviors between Search and Recommendation. In Proceedings of the 47th International ACM SIGIR Conference on Research and Development in Information Retrieval (pp. 1029-1039).
> - Zhao, J. et al. (2025). Unifying Search and Recommendation: A Generative Paradigm Inspired by Information Theory. arXiv preprint arXiv:2504.06714.

---

> ### Author Response · Authors · 2025-11-25
> **Official Comment by Authors**
>
> Dear Reviewer spp9,
>
> We are writing to ask whether our rebuttal has addressed your concerns. We are happy to discuss any remaining issues.

---

> ### Author Response · Authors · 2025-11-26
> **Official Comment by Authors**
>
> Dear Reviewer spp9,
>
> I hope this message finds you well. As the discussion period is nearing its end, I wanted to ensure we have addressed all your concerns satisfactorily. If there are any additional points or feedback you'd like us to consider, please let us know. Your insights are invaluable to us, and we're eager to address any remaining issues to improve out work.
>
> Thank you for your time and effort in reviewing our paper.

---

### Official Review · Reviewer_JC9a · 2025-11-07

**Soundness:** 2
**Presentation:** 3
**Contribution:** 3
**Rating:** 6
**Confidence:** 3

**Summary:**

This paper presents IntSR, a unified generative framework aiming to jointly handle both search and recommendation tasks, as well as their retrieval and ranking subcomponents. The core idea is to model user-item interactions—both search and recommendation—within an autoregressive generative model, distinguishing the task modality through query representations.

**Strengths:**

1. IntSR achieves excellent results in both offline experiments and online A/B tests.

2. The authors conducted comprehensive ablation studies that demonstrate the effectiveness of each design of IntSR.

3. The paper is well-written, and the figures are clear and illustrative.

**Weaknesses:**

1. No direct efficiency experiments. The paper claims computational speed-up but only provides theoretical complexity analysis. It would be much stronger to include a table comparing actual inference latency throughput against baselines.
2. What about scaling? The introduction mentions scalability as a key motivation. How does IntSR's performance scale with model size or data volume? Any experiments or insights on its scaling behavior would be great.
2. Missing discussion of recent work. There's no comparison or discussion with relevant concurrent work, such as [1]. Briefly discussing this would better position the paper.

[1] Shi, Teng, et al. "Unified Generative Search and Recommendation."

**Questions:**

Given the impressive online performance and successful deployment of IntSR, I'm particularly interested in which scenarios the advantages of a unified search-and-recommendation approach are most pronounced？

---

> ### Author Response · Authors · 2025-11-21
> **Official Comment by Authors**
>
> Thank you for the detailed feedback.
>
> **To W1**:
> The setup and results of latency and throughput experiments have been added in the revised submission, line 1010-1034.
>
> QDB increases training throughput by a factor of 4.76x, and increases inference throughput by 1.88x while reducing per-instance latency by 18% on Amazon dataset. In our industrial application, we employ a stream-training paradigm where query tokens are assigned only to the newly added interactions in a user's history. This leads to a much lower overall ratio of query tokens.
> We measured both inference latency (average time to process a single instance) and throughput (instances processed per second) on the same hardware environment (a single NVIDIA H20 GPU with 96 GB memory) with a fixed batch size 64. The results are summarized below:
>
> | Model version | Training throughput (instances/sec) ↑ | Inference latency (ms/instance) ↓ | Inference throughput (instances/sec) ↑ |
> |---|---|---|---|
> | Original module | 21 (1.0×) | 22.5 (1.0×) | 85 (1.0×) |
> | QDB | 100 (4.76×) | 18.5 (0.82×) | 160 (1.88×) |
>
>
> **To W3**:
> For the revised version of our paper, we have rewritten lines 463-480, which comprise the related work.
>
> The modeling paradigm of adopting LLMs to recommendation can be categoried into two types. The first category leverages LLMs as a direct predictors (Wu et al., 2024). Typically, user's historical interaction sequences, profile features are converted into textual inputs (often via task-specific prompts). LLMs are expected to generate recommendation results based on the inputs. The work mentioned by the reviewer falls into this category. However, this approach presents significant implementation challenges in scenarios that require large-scale deployment and low-latency responses (Jia et al., 2025). In contrast, our proposed IntSR follows the another paradigm. This approach focuses on transferring the knowledge and capabilities of LLMs into a specialized recommendation model by using the LLM as a feature extractor and reformulating the task autoregressively.
>
> In our industrial application, which must handle approximately 10,000 user requests per second (QPS) with response times at the millisecond level, we found the second route to be more feasible.
>
>
> **To Q1**:
> Our results indicate that Scenario 2 benefited the most from the unified search-and-recommendation modeling, where IntSR led to a 2.76% relative improvement in Click Through Rate (CTR) over the production baseline.
>
> The performance gains of IntSR are attributed to three main aspects: its generative architecture, the unified modeling of search and recommendation, and the temporal alignment of the candidate set. The magnitude of the improvement in each scenario is closely related to its specific characteristics. Due to the constraints of the double-blind review process, we are currently unable to disclose the specific platform or provide explicit details about the application scenarios. However, we can outline their key characteristics as follows:
>
> - Scenario 1: Features a medium-sized candidate set, analogous to product recommendation. Each item has descriptive metadata. User search activity is relatively sparse, and the candidate set exhibits high dynamism (i.e., additions, removals, and reactivations of items).
> - Scenario 2: Features a large-sized candidate set, also analogous to product recommendation with descriptive metadata for each item. User search activity is abundant, but the candidate set has low dynamism.
> - Scenario 3: Features a small, static candidate set with no user search behavior.
>
> The performance variations can be explained by these characteristics. In Scenario 3, where the candidate set is small, the gains are primarily derived from leveraging larger data volumes and more model parameters (i.e., the scaling law). In Scenario 1, the benefit stems chiefly from the temporal alignment strategy, which effectively handles the high dynamism of the candidate set.
>
> The 2.76% relative CTR lift in Scenario 2 is particularly noteworthy. Previous optimization efforts for this scenario had yielded relative CTR improvements of less than 1%. We attribute this substantial gain to the novel integration of search and recommendation data within a unified framework, as the search data contains explicit user intent that was not previously leveraged in this manner.
>
> Please let us know if any concerns remain unaddressed; we are happy to discuss them.
>
>
> References:
> - Wu, L., Zheng, Z., Qiu, Z., Wang, H., Gu, H., Shen, T., ... & Chen, E. (2024). A survey on large language models for recommendation. World Wide Web, 27(5), 60.
> - Jia et al., 2025. LEARN: Knowledge Adaptation from Large Language Model to Recommendation for Practical Industrial Application. AAAI 25.

---

> ### Author Response · Authors · 2025-11-24
> **Official Comment by Authors**
>
> **To W2**:
>
> The scalability of generative framework in recommendation has been validated by existing studies (e.g., Zhai et al., 2024). The QDB (query driven block) we proposed accelerates any target-aware transformer architecture, without changing its scaling characteristics. We selected one of the three real-world application scenarios mentioned in our submission. This scenario, which involves hundreds of millions of daily active users generating behavioral sequences, provides an ideal testbed for scalability.
>
> We tested the model with varying parameter size and report the results below. The results show a consistent improvement in top1 accuracy as the model size increases, which indicates the model's scalability. In the table below, we only computed QDB's parameters and its forward FLOPs.
>
> | Gflops/example | Parameters | Setting | Top1 accuracy |
> |:---:|:---:|:---|:---:|
> | 7.1 | 0.28B | emb_size=1024, QDBx2 | 0.8151 |
> | 7.6 | 0.27B | emb_size=512, QDBx8 | 0.8217 |
> | 14.3 | 0.55B | emb_size=1024, QDBx4 | 0.8244 |
> | 16.4 | 0.62B | emb_size=768, QDBx8 | 0.8238 |
> | 28.6 | 1.1B | emb_size=1024, QDBx8 | 0.8271 |
>
>
>
> Reference:
> - Zhai, J., Liao, L., Liu, X., Wang, Y., Li, R., Cao, X., ... & Shi, Y. (2024, July). Actions speak louder than words: trillion-parameter sequential transducers for generative recommendations. In Proceedings of the 41st International Conference on Machine Learning (pp. 58484-58509).

---

> > ### Comment · Reviewer_JC9a · 2025-11-25
> >
> > Thanks for your reply. I will keep my score unchanged.

---

> > > ### Author Response · Authors · 2025-11-25
> > >
> > > Thank you for your prompt feedback and for your support of our work and rebuttal.

---

### Author Response · Authors · 2025-11-21
**Brief introduction to the revised PDF**

We thank all reviewers for their time and constructive feedback.

The update version of our submission contains following revisions:

- The validation of the scaling laws has been added in lines 469–495, as suggested by Reviewer JC9a.
- The results of the latency and throughput experiments have been included in Appendix D (lines 1010–1034), as recommended by Reviewer JC9a.
- We have revised and restructured a portion of the Related Work (lines 506–524) based on suggestions from Reviewers JC9a and pYA5.
- Statistical significance is now described when presenting experimental results (lines 343–368 and 378–402), following Reviewer pYA5’s recommendation.
- Explanation of two symbols with subscript are added in lines 214-216 to make the submission self-contained according to the suggestions from reviewer pYA5.
- We have moved the baseline description from the appendix to the main text (see lines 306–326).

---

### Meta-Review · Area_Chair_Qh6d · 2026-01-08

**Summary:**

The paper proposes IntSR, an integrated generative framework designed to unify Search and Recommendation (S&R) tasks. The authors identify query modality (explicit vs. implicit) as the key differentiator and introduce a mechanism to handle these distinct modalities within a single framework. The submission reports impressive results from a large-scale industrial deployment, showing significant gains in GMV, CTR, and accuracy.

The reviewers acknowledged the strong empirical performance and the practical value of a deployed system operating at scale. The problem of unifying S&R is timely, and the reported metrics indicate the system is effective in a production environment. Nevertheless, most reviewers took this manuscript as a borderline paper of ICLR.

The AC is more on the negative side of the borderline because it is hard to treat the integration of search and recommendation as a unique learning problem suitable for the scope of ICLR. This work would be better recognized by the IR & recommendation community if submitted to corresponding conferences like SIGIR/WWW. In addition, the challenges highlighted are not caused by the integration, making the position of the technical contribution unclear.

**Reviewer Concerns:**

More concerns would be addressed.

**Reviewer Scores:**

Unlike changing significantly.

---

### Decision · Program_Chairs · 2026-01-26

Reject